# Mechanisms of DNA Replication and Repair: Insights from the Study of G-Quadruplexes

**DOI:** 10.3390/molecules24193439

**Published:** 2019-09-22

**Authors:** Tracy M. Bryan

**Affiliations:** Children’s Medical Research Institute, University of Sydney, Westmead, NSW 2145, Australia; tbryan@cmri.org.au

**Keywords:** G-quadruplex, DNA replication, DNA structure

## Abstract

G-quadruplexes are four-stranded guanine-rich structures that have been demonstrated to occur across the genome in humans and other organisms. They provide regulatory functions during transcription, translation and immunoglobulin gene rearrangement, but there is also a large amount of evidence that they can present a potent barrier to the DNA replication machinery. This mini-review will summarize recent advances in understanding the many strategies nature has evolved to overcome G-quadruplex-mediated replication blockage, including removal of the structure by helicases or nucleases, or circumventing the deleterious effects on the genome through homologous recombination, alternative end-joining or synthesis re-priming. Paradoxically, G-quadruplexes have also recently been demonstrated to provide a positive role in stimulating the initiation of DNA replication. These recent studies have not only illuminated the many roles and consequences of G-quadruplexes, but have also provided fundamental insights into the general mechanisms of DNA replication and its links with genetic and epigenetic stability.

## 1. Introduction

Guanine-rich DNA has a well-characterized ability to form into four-stranded structures known as G-quadruplexes (G4s), stabilized by hydrogen bonding between a planar association of four guanines [1]. Once thought to be an in vitro curiosity, G-quadruplexes are now known to form throughout genomes in vivo, where they perform positive regulatory roles in addition to potentially having deleterious effects on genome stability. Computational algorithms predict that somewhere between 300,000 and 1.5 million sequences in the human genome are capable of forming into a G-quadruplex [2,3,4], and about 700,000 of these were detected using a sequencing-based method [5]. Sequences with G4-forming potential are enriched in regulatory regions of the genome including promoters and 5′ untranslated regions, and in G-rich repetitive regions such as telomeres [6,7,8,9]. However, an increasing number of proteins have been demonstrated to resolve G4 structures in vitro and in vivo [10,11], so the actual number of G-quadruplexes present at any time is likely to be a much smaller number; chromatin immunoprecipitation with a G4-specific antibody recovered ~10,000 G-quadruplexes from the genome of a human cell line, and confirmed their predicted enrichment in regulatory regions of the genome [12].

G-quadruplexes have many beneficial functions in the genome: they protect telomeres, form binding sites for transcription factors, regulate translation and promote immunoglobulin gene recombination. These roles of G-quadruplexes have been the subject of several reviews [13,14,15]; here we will instead focus on the consequences of G-quadruplex formation for DNA replication. There is a large amount of direct and indirect evidence that G-quadruplexes form an impediment for the DNA replication machinery. The first line of defence for a cell is to deploy one or more helicases or nucleases to remove the G-quadruplex (Figure 1A). If this fails, one of several recovery pathways can minimize the damage: alternative end-joining (mediated by polymerase θ; Figure 1B) [16], homologous recombination (mediated by BRCA1 and BRCA2; Figure 1C) [17,18], or re-priming of synthesis on the other side of the structure (mediated by PrimPol) [19]. Without these back-up mechanisms, the ensuing deletions, recombination and genetic instability can be lethal to the cell. On the other hand, recent data point to a positive role for G-quadruplexes in the initiation step of DNA replication. This mini-review will describe these exciting recent advances in understanding the interplay between G-quadruplexes and DNA replication. These studies have not only revealed the biological effects of these fascinating structures, but have also provided fundamental insights into the mechanisms by which DNA replication is coupled to epigenetic gene regulation and genome stability.

## 2. Evidence for G4-Mediated Replication Stalling

It is well-established that G-quadruplexes stall DNA polymerases in vitro [21], and the inability to bypass these obstacles has now been demonstrated for a range of human replicative and trans-lesion polymerases [22,23,24]. Some of the earliest evidence for the negative effects of G-quadruplexes on DNA replication in vivo came from the analysis of the cellular consequences of small molecules that specifically stabilize these structures. The treatment of immortalized human fibroblasts with the G4-stabilizing molecule RHPS4 resulted in DNA damage foci (marked by the phosphorylated histone γH2AX) exclusively in S-phase cells [25]. This compound particularly impacted replication through telomeres, which are highly prone to G4 formation, resulting in telomere aberrations detected by fluorescence in situ hybridization (FISH) as “doublets” [25]; these aberrations have been demonstrated to result from telomere replication defects, and have become known as a “fragile telomere” phenotype [26]. An unrelated G4-stabilizing compound, 360A, also preferentially targets telomeres, and was seen to cause a fragile telomere phenotype specifically on the lagging strand, which is predicted to be the G-rich strand at telomeres [27]. On the other hand, the molecule pyridostatin causes DNA damage across the genome of human cells [28]. Approximately half of the γH2AX marks occurred in S phase and were sensitive to the DNA replication inhibitor aphidicolin, whereas the remainder occurred in other phases of the cell cycle and were sensitive to the inhibition of transcription [28]. These data illustrate that while blocking replication is not the only cause of DNA damage in human cells with over-stabilized G-quadruplexes, it is a major contributor.

Experiments involving the depletion of one of the many helicases known to resolve G-quadruplexes [10] have also revealed the impact of the inappropriate stabilization of these structures on DNA replication. BLM-deficient mouse fibroblasts demonstrate an increase in the number of nuclear foci detected with a G4-specific antibody [29], indicating that the well-documented ability of the BLM helicase to unwind G-quadruplexes in vitro [30,31] is recapitulated in vivo. BLM deficiency in mouse cells leads to an increase in fragile telomeres [26,32,33]; these predominantly occur on the lagging strand, providing indirect evidence that it is the ability of BLM to unwind G-quadruplexes that prevents replication defects. Similarly, depletion of the related helicase WRN caused replication defects specifically on the lagging strand of human telomeres [34,35]. Deletion of the helicase RTel or the nuclease DNA2, both of which have been demonstrated to remove G-quadruplexes in vitro [36,37], also caused an increase in fragile telomeres in mouse cells, which was greatly exacerbated in the presence of G4-stabilizing compounds [33,38,39].

Direct evidence for the impact of G-quadruplexes on replication in vivo has come from experiments using single-molecule analysis of replicating DNA, in which DNA molecules are stretched on microscope slides and the patterns of nucleotide incorporation are detected using FISH [40]. This technique demonstrated that the speed of the replication fork travelling through telomeres was reduced in BLM-deficient mouse cells relative to controls, and this was exacerbated in the presence of the G4-stabilizing compound PhenDC3, providing evidence that BLM affects replication through its ability to unwind G-quadruplexes [29]. Similar results have been obtained for chicken cells lacking the helicase FANCJ [41].

A surprising recent addition to the family of proteins that can remove telomeric G-quadruplexes is the human CST (CTC1–STN1–TEN1) complex, an RPA-like complex that binds single-stranded DNA. Mammalian CST has wide-ranging roles in DNA replication at telomeres and across the genome; it facilitates replication restart after fork stalling, recruits polymerase α to mediate fill-in synthesis of the telomeric C-strand, and regulates the addition of telomere repeats by the ribonucleoprotein telomerase [42]. It has recently been demonstrated that human CST can bind and unwind G-quadruplexes, both in vitro and in vivo [43,44]. The depletion of CST resulted in fragile telomeres and the loss of the C-strand of telomeres, in a manner that was synergistic with treatment with a G-quadruplex stabilizer [43]. The role of CST in overcoming G4-mediated blocks to replication may involve its recruitment of polymerase α [43] or its previously documented ability to recruit RAD51 to sites of replication stress [45], in addition to a direct role in resolving G-quadruplexes.

There is also recent evidence that the chromatin-remodelling protein ATRX plays a role in removal of G-quadruplexes to facilitate DNA replication in mammals. ATRX is mutated both in patients with the X-linked alpha thalassemia mental retardation syndrome (ATR-X syndrome), and in cancers that use a non-telomerase mechanism for lengthening their telomeres [46]. ATRX appears to coordinate a multitude of cellular processes, with roles in gene expression, preventing mitotic spindle defects, regulating nucleosome density and affecting chromatin looping, in addition to its most well-characterized role in the deposition of histone variant H3.3 at telomeres and heterochromatic regions [46]. To what extent these functions are linked by a common molecular mechanism is unknown, but recent evidence suggests that ATRX may have a key role in facilitating DNA replication by promoting the unwinding of G-quadruplexes. ATRX binds to telomeres, other tandem repeats, and CpG-rich regions across the human genome, and ~50% of these sites are predicted to form into G-quadruplexes [47]. ATRX can bind G-quadruplexes in vitro [47]; it has not yet been shown to directly resolve them [48], but its depletion leads to an increase in the antibody detection of G-quadruplexes in human cells [49]. An involvement of ATRX in replication progression was suggested by the increased susceptibility of ATRX-deficient mouse and human cells to the replication blockers hydroxyurea and aphidicolin [50,51,52], and it has also been observed that ATRX depletion causes an increase in stalled replication forks and fragile telomeres [48,51,52,53]. ATRX-deficient cells are highly sensitive to proliferation defects caused by G4-stabilizing ligands [49,50,54], supporting a model in which ATRX promotes the resolution of G-quadruplexes at replication forks, helping to overcome replication stalling in regions of the genome that are difficult to replicate [55].

## 3. Downstream Effects of Stalled Replication: Genome Deletions and Rearrangements

Replication fork blocks are well known to result in a range of deleterious events such as double-strand breaks (DSBs), deletions and copy number alterations. The molecular events leading to these outcomes are an active area of investigation [56], and the study of G-quadruplexes has contributed to discovery of global pathways by which cells respond to replication stress.

One of the first examples of the genome instability caused by G-quadruplexes came from analysis of the FANCJ helicase and its homologue in the worm *Caenorhabditis elegans*, DOG1. Worms lacking DOG1 and cells from human Fanconi anemia patients with mutations in *FANCJ* carry a large number of genomic deletions, which are enriched in regions of the genome with G4-forming potential [57,58,59]. The construction of a selectable marker containing G-quadruplexes in the *C. elegans* genome enabled a detailed molecular analysis of the nature of the mutations, and the discovery of an alternative DSB repair pathway [16]. The deletions had a defined size range (50–300 bp), their 3′ ends lined up with the predicted boundary of the G-quadruplex, and they often had 1 nt of microhomology at their breakpoint junctions. Remarkably, deletion of polymerase θ resulted in much larger deletions of >10 kb, indicating that polymerase θ is needed to mitigate the damage resulting from the lesion. The authors proposed a model in which polymerase θ mediates the end joining and gap filling of a DSB arising from replication fork blockage (Figure 1B)—a process they termed theta-mediated end joining (TMEJ) [16]; this process is also now known as microhomology-mediated end joining (MMEJ) or alternative non-homologous end-joining (Alt-NHEJ) [60]. Furthermore, analysis of the inheritance of different deletion alleles over multiple cell divisions in the worm revealed that the persistence of the G-quadruplex structure through multiple mitotic divisions was responsible for converting the original single-stranded gap opposite the G4 into a subsequent DSB, to be acted on by TMEJ [61].

Another rescue pathway for mitigating damage from stalled replication forks is homologous recombination (HR), which can restart forks blocked by a replication barrier [62]. The helicase Pif1 is essential for facilitating replication through G4-containing sequences; its absence leads to genomic instability and gross chromosomal rearrangements in the yeast *Saccharomyces cerevisiae* [63,64,65]. It has been directly demonstrated that G4 sequences cause slowing of the replication fork in the absence of Pif1 or its *Schizosaccharomyces pombe* homologue Pfh1 [66,67,68]. Analysis of the mechanism of genomic rearrangements occurring around a G4-prone minisatellite sequence revealed X-shaped recombination intermediates, and both these and the genomic rearrangements depended on the presence of HR proteins RAD51 and RAD52 [20]. These data support a model in which the replication blockage caused by G-quadruplexes is rescued by HR-mediated fork restart and template switching around the region of the blockage (Figure 1C). Furthermore, human Pif1 has recently been shown to directly participate in the HR process at stabilized G-quadruplexes through an interaction with the HR protein BRCA1 [69]. RAD51 is also necessary for the appearance of fragile telomeres in human cells treated with G4-stabilizing compound 360A [27], suggesting that the gaps in telomere FISH signal that are diagnostic of these structures are a result of homologous recombination-mediated repair around the G-quadruplex obstruction.

The involvement of HR in recovery from G4-mediated genome damage has recently been reinforced by studies in which G4-stabilizing compounds exhibited specific toxicity against human cells lacking BRCA1, BRCA2 or RAD51 [17,18,54,70]. Depletion of these proteins caused replication defects including lagging-strand fragile telomeres and shortened fibres in single-molecule assays, and these effects were increased after treatment with G4-stabilizing compounds [17,18], implying that HR was needed for defects arising from G4-mediated replication blockage (Figure 1C). These studies raise the exciting possibility that G4-stabilizing molecules may prove to be an effective cancer therapy in patients with defects in HR proteins such as BRCA1 and BRCA2, providing a “synthetic lethal” approach to cancer therapy for these patients.

In addition to presenting a direct impediment to DNA replication, G-quadruplexes may harbor DNA lesions that themselves block replication; for example, G-rich sequences are particularly prone to oxidative damage [71,72]. Replication stalling and genome damage caused by oxidative lesions can be mitigated by the base excision repair (BER) pathway; in particular, the NEIL3 DNA glycosylase can remove certain oxidized guanine lesions from G-quadruplex DNA [73,74]. This ability may contribute to the ability of NEIL3 and BER to promote the replication of telomeric DNA, since the depletion of NEIL3 in human cells leads to an increase in fragile telomeres [75].

Together, the above studies illustrate that one of several DNA repair pathways (TMEJ, HR, BER) can be deployed to overcome replicative DNA damage caused by G-quadruplexes and their associated DNA lesions.

## 4. Downstream Effects of Stalled Replication: Changes in Epigenetic Gene Regulation

The effects of G-quadruplexes need to be considered in the context of DNA organized into nucleosomes, in which DNA is wrapped around a core of histone proteins that carry post-translational modifications dictating the identity of the surrounding chromatin (Figure 2). After DNA has been replicated, it is vital for cellular identity that the pattern of histone modifications present on the parental strand is re-established on the newly replicated daughter strands. Nucleosomes are disassembled ahead of the DNA replication fork, and recycled histones are combined with new histones and deposited onto each of the daughter strands. It is thought that the posttranslational marks of the parental histones influence acquisition of the same marks by new histones (Figure 2) [76,77]. Replication stress is known to perturb histone recycling and re-establishment of chromatin marks [78], and there is a growing body of evidence that G-quadruplexes can contribute to epigenetic deregulation after DNA replication.

Many of the studies linking G-quadruplexes with post-replicative epigenetic changes have been carried out in the genetically tractable chicken cell line DT40. G-quadruplexes have been shown to form at several gene loci in DT40, including the β-globin gene and those encoding cell surface markers CD72 and BU-1 [79,80]. Loss of proteins known to process G-quadruplexes, including FANCJ, BLM and WRN, or loss of the trans-lesion polymerase REV1, resulted in changes in histone marks deposited in the vicinity of these G-quadruplexes and changes in expression of the genes, in a manner dependent on the presence and orientation of the G-quadruplex [79,80,81]. Loss of FANCJ in DT40 cells also resulted in global changes in chromatin compaction and an increase in single-stranded DNA across the genome—effects which were recapitulated by the G-quadruplex-stabilizing ligand telomestatin [41].

These observations suggest a model in which failure to replicate G-quadruplexes leads to post-replicative gaps that are later replicated in a manner uncoupled from the replication fork and the supply of parental histones, resulting in the incorporation of new histones without the parental chromatin marks (Figure 2). The requirement for processive DNA synthesis for the maintenance of epigenetic memory was supported by the observation that perturbing replication with hydroxyurea or aphidicolin caused very similar epigenetic changes, and in the case of the BU-1 locus these were partially dependent on the presence of the G-quadruplex [41,82]. G-quadruplex-stabilizing ligands caused gene expression changes at these loci in a manner synergistic with hydroxyurea, and the effects on gene expression remained even after removal of the compound [82,83], demonstrating that they were the result of stable epigenetic modification rather than transient impacts on transcription or translation.

The requirement for FANCJ, BLM and WRN in removing G-quadruplexes in order to maintain chromatin structure can be explained by their known G4-helicase activities, but the role of the polymerase REV1 in this process is not fully understood. REV1 is known to coordinate other polymerases in the DNA repair process known as trans-lesion synthesis [84], and both its region of interaction with these polymerases and its own catalytic activity are required for its ability to promote the replication of a G4-containing template [79]. REV1 is a highly specialized polymerase; it incorporates only cytosines, with the identity of the incoming nucleotide being dictated by the enzyme itself [85,86]. It is therefore possible that REV1 is needed to incorporate cytosines opposite newly unwound guanines while copying past a G-quadruplex.

The recently identified polymerase DNA-directed primase/polymerase (PrimPol) is also vital for avoiding effects of G-quadruplexes and other secondary structures on epigenetic stability. PrimPol can bind to G-quadruplexes, but does not unwind them; instead, it re-primes synthesis immediately 3′ of an intervening G-quadruplex, allowing replicative bypass of the obstruction [19]. Consequently, loss of PrimPol causes the same epigenetic changes at the Bu-1 locus of DT40 cells as loss of FANCJ or REV1. The ability of PrimPol to bypass replicative lesions is not limited to G-quadruplexes; it also counteracts epigenetic instability caused by replication across a (GAA)_n_ repeat, which is known to form triplex DNA [87]. The effect of losing PrimPol on epigenetic defects caused by G4 or (GAA)_n_ was counteracted by the overexpression of RNase H; this suggests that the formation of RNA–DNA hybrids, or R-loops, on the strand opposite the obstruction can increase the probability of the secondary structure impeding replication (Figure 2). Loss of PrimPol resulted in increased R-loop formation across the genome of both chicken and human cells, particularly in regions predicted to form triplexes or G-quadruplexes, demonstrating that this polymerase has a genome-wide role in suppressing R-loops associated with secondary structure formation on the opposite strand [87].

There is an increasing body of evidence demonstrating that G-quadruplexes form in regulatory regions of the genome, and that their presence in promoters and 5′ untranslated regions of genes directly regulates gene transcription [15]. The studies summarized above demonstrate that G-quadruplexes can also influence gene expression through post-replicative effects on chromatin remodelling. In wild-type cells, many helicases and polymerases collaborate to counteract this effect; this may be why a vast majority of G-quadruplexes in the human genome occur in nucleosome-free regions [12].

## 5. Role for G-Quadruplexes in Initiation of DNA Replication

The large amount of literature documenting evidence for G4-mediated interference with DNA replication gives the overall impression that these structures are inherently deleterious. However, it should be remembered that most examples of the negative effects of G4 on DNA replication occur in pathological situations (e.g., in the absence of helicases or in the presence of G4-stabilizing compounds). In wild-type cells under non-perturbed conditions, it is likely that the plethora of G4-unwinding proteins is able to counteract any negative effects of G-quadruplexes on DNA synthesis. Moreover, there is also recent evidence to suggest that G-quadruplexes play a necessary and positive role in the initiation of DNA replication.

The genome-wide mapping of metazoan replication origins revealed that a majority (70%–90%) of origins are preceded by a G-rich sequence that has G4-forming ability, known as an Origin G-rich Repeated Element (OGRE) (Figure 3) [88,89,90,91]. The OGRE lies 250–300 bp upstream of the replication initiation site, in a nucleosome-free region [90]. Experimental manipulation of OGREs located at two origins in chicken cells provided the first evidence that their G4-forming ability is necessary for optimal origin activity [92]. Indeed, the combined stability of different G4s within one of these sequences correlated strongly with the activity of its associated origin of replication [92]. In apparent contradiction to these data, however, allele-specific analysis of a subset of origins in human cells found no correlation between origin activity and the presence of a predicted G4 sequence [93].

A recent elegant study used a number of independent approaches to resolve this apparent inconsistency, demonstrating that human replication origins fall into two main classes: those that are regulated by G-quadruplex formation, and those that are instead primarily activated by transcription [94]. This study provided the first definitive biophysical evidence for G4 formation by OGRE sequences, and used CRISPR-mediated genome editing to show that a reduced capacity for G4 formation resulted in a reduction in origin activity. Treatment of human cells with a G4-stabilizing compound resulted in increased activity of some origins, and decreased activity of others. The latter were highly enriched for origins lacking G4-forming motifs, and were associated with actively transcribing promoters [94]. The authors concluded that the presence of G4 structures may promote origin activation in intergenic areas, which cannot benefit from the known origin-promoting effects of active transcription.

The mechanism of G4-mediated origin activation has not been fully elucidated, but may involve the known G4-binding capability of a number of replication-related proteins, including the origin recognition complex (ORC) [95] or MTBP—a protein that is needed for assembly of the CMG (CDC45–MCM–GINS) complex during origin firing [96,97]. In support of a role for G4 in origin firing rather than origin licensing, G4-forming OGRE sequences competed with the chromatin loading of CMG component CDC45, but not of ORC, in *Xenopus* extracts [94]. Overlay of human genome-wide G4-sequencing data [5] with the location of both active and dormant origins [98] suggested that G4 structures form preferentially at firing origins, also consistent with a role for G4 in the efficiency of origin firing. It has been postulated that the role of OGRE G4 sequences may be related to the exclusion of nucleosomes, the recruitment of DNA-unwinding helicases, or the presence of single-stranded DNA in the strand opposite the G4 [94,99], but there is no direct evidence for these hypotheses as of yet.

## 6. Future Directions: Therapeutic Possibilities

It is apparent that the genome exists in conformations other than just duplex DNA. The study of the genomic locations and biological consequences of G-quadruplex DNA is revealing fascinating insights into the regulation of DNA replication that are likely generalizable to other secondary structures and causes of replication stress. It has revealed new modes of recovery from replicative DNA damage, and novel mechanisms of the initiation of DNA replication at replication origins. However, these new insights are not only of academic interest; they also have far-reaching implications for understanding human disease. Almost all of the proteins that are involved in processing G-quadruplexes are linked to one or more diseases. FANCJ is one of the set of 19 proteins that are mutated in Fanconi anemia—a genetic disease that results in sensitivity to genetic damage and predisposes patients to bone marrow failure and cancer development. Components of the CST complex are mutated in Coats plus syndrome—an inherited condition characterized by an eye disorder, abnormalities of the brain, bones and gastrointestinal system, and other features including anemia. BLM is mutated in Bloom’s syndrome—an inherited disorder characterized by short stature, UV sensitivity, and a greatly increased risk of cancer. Mutations in WRN cause Werner’s syndrome, a premature aging syndrome that also imparts an increased risk of cancer. These disorders have some overlapping features, but are also very divergent. It will be interesting to determine which of these clinical features are contributed by the shared function of these proteins in facilitating DNA replication through structured DNA, and what biochemical functions of the proteins lead to other clinical features. Such insights may eventually lead to improved therapeutic options for patients with these inherited disorders; for example, small molecules that lower the stability of G-quadruplexes may substitute for the G4-resolving ability of each of these proteins, improving any patient symptoms that result from G4-mediated genome instability (Figure 4).

Furthermore, G4-mediated replication problems are also implicated in cancer. The genetic instability caused by G-quadruplexes and other structures is likely a driving force in cancer, but may also be harnessed to specifically kill cancer cells that have deficiencies in the recovery pathways to deal with structured DNA. The approach of combining the pharmacological targeting of one pathway together with the genetic targeting of a cooperating pathway is termed a “synthetic lethal” approach to cancer therapy (Figure 4), and is exemplified by the enhanced susceptibility of cancer cells lacking HR proteins BRCA1 and BRCA2 to death caused by G4-stabilizing ligands [17,70]. These findings have resulted in the molecule CX-5461 being one of the first G4-stabilizing compounds to enter clinical trials in humans, in patients with germline *BRCA1* or *BRCA2* mutations [18,100]. Recently, a genome-wide shRNA screen was performed in cells treated with pyridostatin or PhenDC3 in order to systematically screen for other potential genetic susceptibilities to G4-stabilizing molecules. This resulted in the identification of 50 sensitizing genes that are known to be somatically mutated in human cancers [54]. These included *BRCA1* and *BRCA2* and their interacting partners *PALB2* and *BAP1*, and a cluster of chromatin modifiers including *SMARCA4*, *SMARCB1* and *SMARCE1*. Consistent with its role in mitigating DNA damage caused by inappropriate G4 stabilization, the gene encoding Pol θ (POLQ), which is also mutated in human cancers (https://cancer.sanger.ac.uk/census) also sensitized cells to pyridostatin treatment [54]. This raises the exciting possibility that a large number of human cancers may be susceptible to cell death induced by G4-stabilizing compounds.

Even in cancers without mutations in G4-sensitizing genes, a synthetic lethal approach could include the combination of G4-stabilizing compounds with the pharmacological targeting of proteins needed for recovery from G4-mediated genomic damage. For example, pyridostatin acts synergistically with NU7441, an inhibitor of the DNA-PK kinase that is crucial for nonhomologous end-joining repair of DNA DSBs [70], and with an inhibitor of USP1, which deubiquitinates a protein in the Fanconi anemia DNA repair pathway [54]. Many of the genes found to be G4 sensitizers in the genome-wide screen mentioned above are considered “druggable” [54], including *BRCA1*, making this a promising avenue for the further testing of drug combinations.

Cancers may also harbor mutations in proteins that are needed for the removal of G-quadruplexes in the genome, such as ATRX, which is mutated in a subset of human cancers that use a non-telomerase mechanism for the elongation of telomeres known as ALT (alternative lengthening of telomeres) [101]. Cells lacking ATRX are also particularly sensitive to cell death caused by G4-stabilizing molecules [49,50,54], raising the possibility that these molecules will be a useful therapy for ALT cancers. The finding that G4-specific antibodies detect more G4 in the genomes of cancer cells compared to normal human cells [12,102] suggests that non-ALT cancers also commonly harbor defects in proteins required for normal G-quadruplex regulation. Such proteins may be a fruitful further avenue for synthetic lethal cancer therapies. It is therefore likely that, after decades of biochemical and biophysical examination of G-quadruplexes, we are now moving into the era of translation of this fundamental understanding into human clinical trials.

## Figures and Tables

**Figure 1 molecules-24-03439-f001:**
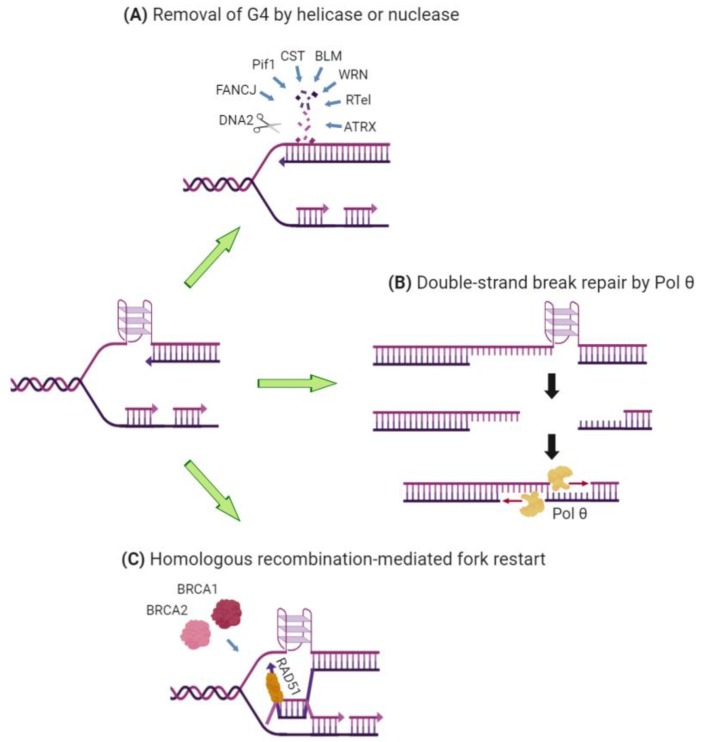
Cellular strategies to minimize genomic damage arising from G-quadruplex (G4)-mediated replication stalling. (**A**) A large number of helicases and nucleases exist to remove G-quadruplexes; it is not yet clear why so many proteins with this function are needed. (**B**) In the absence of FANCJ, deletions occur in the region surrounding G-quadruplexes. The size of these deletions is kept in check by the end-joining and gap-filling activity of polymerase θ [16]. (**C**) In the absence of Pif1, homologous recombination mediated by RAD51, BRCA1, and BRCA2 can lead to genomic rearrangements [20]. *Figure created with BioRender.com*.

**Figure 2 molecules-24-03439-f002:**
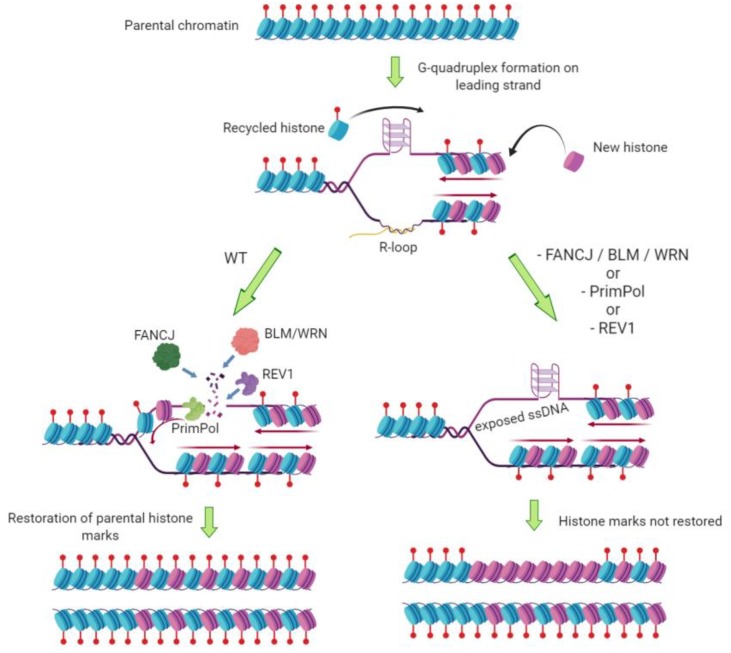
Model for the mechanism of G4-mediated changes to histone modifications and gene expression (adapted from [19,79]). G-quadruplex formation at the replication fork can block replication (here depicted with the G4 on the leading strand, but it may also block lagging strand synthesis). (**Left panel**): FANCJ, BLM or WRN helicases can remove the G-quadruplex, and DNA synthesis past the impediment can be promoted by REV1 or PrimPol. (**Right panel**): in the absence of any of these pathways, an extended region of single-stranded DNA template may accumulate. The subsequent DNA synthesis needed to fill this gap would be uncoupled from the incorporation of parental histones into the newly replicated DNA strand, resulting in a loss of the restoration of parental histone marks. *Figure created with BioRender.com*.

**Figure 3 molecules-24-03439-f003:**
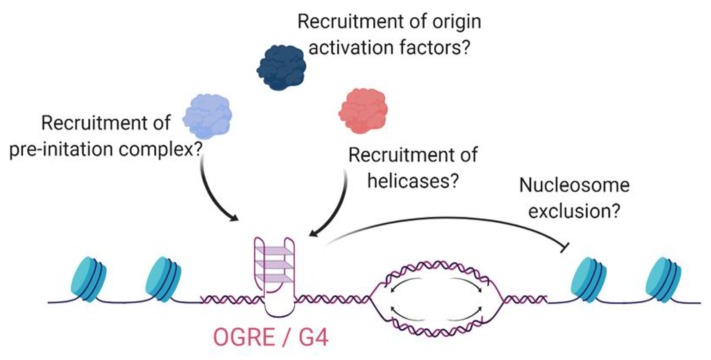
Promotion of replication origin firing by a G4-forming sequence. The Origin G-rich Repeated Element (OGRE) lies 250–300 bp upstream of the replication initiation site, in a nucleosome-free region, and is needed for optimal origin activity of a subset of vertebrate replication origins [90,92,94]. *Figure created with BioRender.com*.

**Figure 4 molecules-24-03439-f004:**
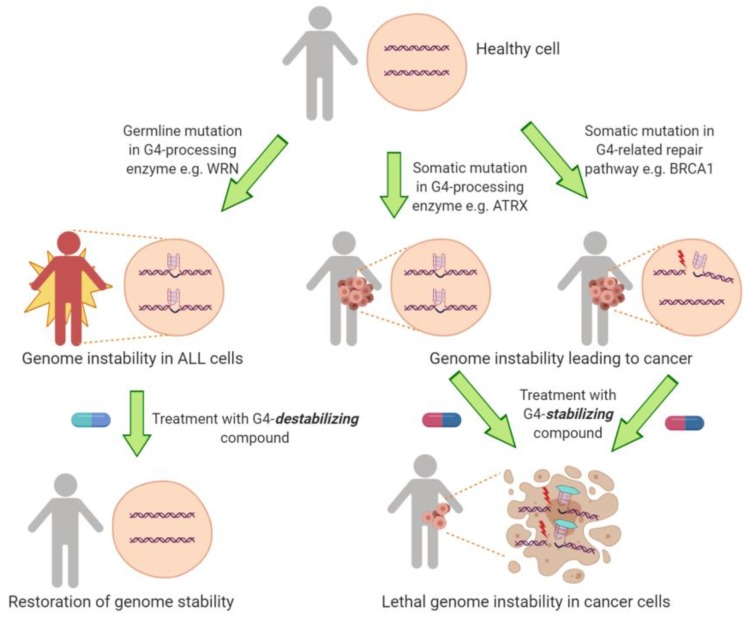
Potential therapeutic applications of G4 effects on the genome. Germline mutations in G4 helicases such as WRN or BLM lead to genomic instability throughout the body, which could be counteracted with G4-destabilizing compounds. Cancer-associated mutations in pathways that deal with G4 could form the basis of synthetic lethal approaches to cancer treatment. *Figure created with BioRender.com*.

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
