# Peer review of "Mechanisms of DNA Replication and Repair: Insights from the Study of G-Quadruplexes"

_molecules, 2019, doi:10.3390/molecules24193439_

Round 1

Reviewer 1 Report

Review attached.

Reviewer 2 Report

In this mini-review, Tracy Bran reviewed the recent progress of DNA replication and repair in the context of quadruplex DNA. The author summarized the evidence of quadruplex mediated replication stalling in intro and in vivo, and the genomic and epigenetic outcomes of such events. A new role of G4 DNA in DNA replication initiation was also reviewed. In general, this is a very well-written, high quality review article that can be published after only minor revision. The figures present the mechanisms very well.

I only have a few suggestions that will be helpful for the fullness of the review.

The author may highlight the potential clinical application of targeting G4 by reviewing these studies in a separate subtitle. It can be on G4-inducing agents, or inhibitors of G4 processing enzymes, i.e. POLQ inhibitor, WRN and BLM inhibitors. The Shieldin complex was recently discovered. which structurally resembles to the telomere CST complex and Shelterin complex. Shieldin and CST complexes counteract DSB resection and prevent genome instability (Nature. 2018 Aug;560(7716):117-121; Nature. 2018 Aug; 560(7716): 112–116.). May be the two complexes have quadruplex resolving capacity, like Shelterin complex. A quick search seems to be so (Mammalian CST averts replication failure by preventing G-quadruplex accumulation. NAR, 2019). The author should also mention the studies on the NEIL3 DNA glycosylases. The enzyme directly acts on G-DNA (Nucleic acids research 43 (8), 4039-4054; JBC 288 (38), 27263-27272), some early evidence of direct BER at G4, and has a role in telomere repair during S phase (Cell reports 20 (9), 2044-2056). May be there’s a link.

Reviewer 3 Report

The review by Tracy Bryan is written in a very elegant and comprehensive manner. The Figures are beautiful and the text provided a nice summery of the current literature. One minor comment I have is that the major focus of the paper is what the consequences of G4s in humans are. Therefore I would suggest to inlcude this in the title and abstract to avoid misunderstanding.

Also I would be more precise in the future direction part, what the field speculates how changes in G4s during disease stages can be used as therpeutical markers for replication defects and maybe even as markers/indicators for genome stabiliy/cancer.

other than this, it is a beautiful manuscript. congratulations
